# $CO_x$ hydrogenation to methanol and other hydrocarbons under mild conditions with $Mo_3S_4$@ZSM-5

Gui Liu[1], Pengfei Liu[2], Deming Meng[1], Taotao Zhao[1], Xiaofeng Qian[1], Qiang He[1], Xuefeng Guo[1], Jizhen Qi[3], Luming Peng[1], Nianhua Xue[1], Yan Zhu[1], Jingyuan Ma[4] ✉, Qiang Wang[2] ✉, Xi Liu[5] ✉, Liwei Chen[3,5] & Weiping Ding[1] ✉

The hydrogenation of $CO_2$ or CO to single organic product has received widespread attentions. Here we show a highly efficient and selective catalyst, $Mo_3S_4$@ions-ZSM-5, with molybdenum sulfide clusters ($[Mo_3S_4]^{n+}$) confined in zeolitic cages of ZSM-5 molecular sieve for the reactions. Using continuous fixed bed reactor, for $CO_2$ hydrogenation to methanol, the catalyst $Mo_3S_4$@NaZSM-5 shows methanol selectivity larger than 98% at 10.2% of carbon dioxide conversion at 180 °C and maintains the catalytic performance without any degeneration during continuous reaction of 1000 h. For CO hydrogenation, the catalyst $Mo_3S_4$@HZSM-5 exhibits a selectivity to $C_2$ and $C_3$ hydrocarbons stably larger than 98% in organics at 260 °C. The structure of the catalysts and the mechanism of $CO_x$ hydrogenation over the catalysts are fully characterized experimentally and theoretically. Based on the results, we envision that the $Mo_3S_4$@ions-ZSM-5 catalysts display the importance of active clusters surrounded by permeable materials as mesocatalysts for discovery of new reactions.

Methanol is one of the most important commodity chemicals and usually prepared in industry by syngas conversion over Cu/ZnO/Al2O3 catalysts at elevated temperatures (230-300 °C) and pressures (5-10 MPa)[1,2]. As an alternative, the hydrogenation of $CO_2$ is also an effective route to methanol synthesis and, as shown in Eqs. (1)-(3), the utilization of chemical energy contained in feed gases of the two routes is in fact similar. Although the hydrogenation of $CO_2$ to methanol, at present time, seems to face some challenges in catalyst activity, selectivity and stability, the intensive study on the reaction would provide a more efficient route for industrial production of methanol and utilization of $CO_2$[3,4]. For the process, however, the side reaction of reverse water-gas shift (RWGS, Eq. 3), is almost

unavoidable and, the higher the temperature, the more serious the RWGS reaction, due to its endothermic nature. It not only lowers selectivity toward methanol, but also intensifies the sintering of the active phase at higher temperature and reduce the stability of the catalyst, due to the by-product water[5]. How to avoid the occurrence of RWGS reaction has been a troublesome question for a long time.

$$CO_2 + 3H_2 \rightarrow CH_3OH(g) + H_2O(g) \quad \triangle H = -49.5 kJ/mol \quad (1)$$

$$CO + 2H_2 \rightarrow CH_3OH(g) \quad \triangle H = -90.1 kJ/mol \quad (2)$$

[1]Key Lab of Mesoscopic Chemistry, School of Chemistry and Chemical Engineering, Nanjing University, Nanjing, China. [2]Department of Applied Chemistry, School of Chemistry and Molecular Engineering, Nanjing Tech University, Nanjing, China. [3]i-Lab, CAS Centre for Excellence in Nanoscience, Suzhou Institute of Nano-Tech and Nano-Bionics, Chinese Academy of Sciences, Suzhou, PR China. [4]Shanghai Synchrotron Radiation Facility, Pudong New District, Shanghai, China. [5]School of Chemistry and Chemical, In-situ Centre for Physical Sciences, Frontiers Science Centre for Transformative Molecules, Shanghai Jiao Tong University, Shanghai, PR China. ✉e-mail: majingyuan@zjlab.org.cn; wangqiang@njtech.edu.cn; liuxi@sjtu.edu.cn; dingwp@nju.edu.cn

$$CO_2 + H_2 \rightarrow CO + H_2O \quad \triangle H = 41.1 kJ/mol \qquad (3)$$

$$CO_2 + 4H_2 \rightarrow CH_4 + 2H_2O \quad \triangle H = -164.9 kJ/mol \qquad (4)$$

The thermodynamic calculation on reactions 1, 3, and 4 with mixture of $CO_2/3H_2$ as feed gases indicates the RWGS reaction is negligible at temperatures lower than 180 °C, as shown in Supplementary Fig. 1, and the formation of $CH_4$ has large advantages in thermodynamics, especially at low temperatures. For products of oxygenates from $CO_2$ hydrogenation, the catalyst must be active for cleavage of the first C-O bond but inactive for the second C-O bond of $CO_2$. It is a great challenge to develop such a catalyst effective for $CO_2$ hydrogenation at temperatures lower than 180 °C.

In other aspects, the hydrogenations of $CO_2$ and CO have complicated multi-relation. The product distribution of CO hydrogenation (Fischer-Tropsch synthesis, FTS) generally follows the famous Anderson-Schulz-Flory (ASF) law. Breaking through the limitations of ASF model for FTS reactions to obtain single or several products has been of particular importance and long term desired[6–8]. To synthesize methanol via $CO_2$ hydrogenation reaction in near exclusive selectivity is very meaningful and the mechanism elucidated can offer basises to develop efficient catalysts for synthesis of low hydrocarbons in specially high selectivity.

In recent years, the catalysts active for $CO_2$ hydrogenation have been investigated widely and the most valued catalysts would be Cu/ZnO/$Al_2O_3$[9–12] and indium oxide-based catalysts[3,5,13]. For Cu/ZnO/$Al_2O_3$ catalyst, the main problem would be the unavoidable RWGS reaction which lowers methanol selectivity. Other copper-based catalysts such as Cu/ZnO[14], Cu/$CeO_x$/$TiO_2$[1] and Cu/$ZrO_2$[15] often give the selectivity to methanol less than 70% in hydrogenation of $CO_2$. Indium oxide-based catalysts have been studied intensively and have shown great potential in the hydrogenation of $CO_2$ to methanol. A 7% conversion of $CO_2$ and 40% selectivity toward methanol can be achieved with pure $In_2O_3$ catalyst under the reaction conditions of 330 °C and 5 MPa[16]. A selectivity to methanol > 99% over $In_2O_3$/$ZrO_2$ catalyst has been reported with $CO_2$ conversion of only ~5%[4]. In addition to copper and indium oxide-based catalysts, ZnO-$ZrO_2$ composite oxide catalyst has also made great progress in the hydrogenation of $CO_2$ to methanol, which achieved 86% selectivity toward methanol and more than 10% conversion of $CO_2$ at 5 MPa and 320 °C[17].

We have been impressed for a long time by unique catalytic properties of the composite composed of ZSM-5 zeolite and its intracrystalline tough clusters of metal compounds, such as $MoN_x$@ZSM-5[18], $Pt_x$@ZSM-5[19], and Iglesia's work on $MoC_x$@ZSM-5[20] and $WC_x$@ZSM-5[21]. The strong regulatory effect of ZSM-5 on the confined clusters and the effect of cooperations among the clusters, the zeoliti acidic sites and the zeolitic porosity on conversions of reactive molecules lead to the catalytic property of the composite unpredictable and extremely interesting. In this work, we report a highly efficient catalyst combined $[Mo_3S_4]^{n+}$ clusters as active centers and porous NaZSM-5 as confining or peripheral framework for the hydrogenation of $CO_2$ to methanol in fixed bed reactor (Mo: 3 wt%, $Mo_3S_4$: 4.3 wt%). Interestingly, our current study is related to recent report by Wang and Deng[22], where they have reported a catalyst of few-layer $MoS_2$ with sulfur vacancy for the hydrogenation of $CO_2$ to methanol, showing similar catalytic performance but in different mechanism. The S/Mo ratio of current $Mo_3S_4$@NaZSM-5 catalyst is much lower than that of $MoS_2$. More interestingly, when the $Mo_3S_4$@HZSM-5 (with proton as movable equilibrium ions) catalyst is used for syngas (2CO/$H_2$) conversion, noteworthy that it exhibits a very stable selectivity to $C_2$ and $C_3$ hydrocarbons > 98% at 260 °C, or ~ 90% to $C_2$-$C_4$ hydrocarbons at 400 °C with conversion of CO ~ 20%, indicative of the rich catalytic properties of the catalysts $Mo_3S_4$@ions-ZSM-5 for $CO_x$ conversion.

## Results

### Structure and physicochemical properties of the catalysts

The $Mo_3S_4$@NaZSM-5 catalyst with $[Mo_3S_4]^{n+}$ positioned in cages of ZSM-5 was synthesized by two-step mechanical mixing and calcination method, followed by neutralization of the zoelitic acidic sites in 0.05 M aqueous solution of NaOH, as schematically shown in Fig. 1a (details see Methods section). The structure of a ZSM-5 cage (Fig. 1b) containing a $[Mo_3S_4]^{n+}$ cluster (Fig. 1c) was calculated and optimized using DFT method and the result was depicted in Fig. 1d, which is highly reliable to be described as $Mo_3S_4$@NaZSM-5, in accordance with the results of X-ray absorption fine structure (XAFS) and X-ray photoelectron spectroscopy (XPS) characterizations (Fig. 3).

The microstructures of the fresh and spent $Mo_3S_4$@NaZSM-5 catalysts were observed using Probe-Corrected Scanning Transmission Electron Microscope (STEM). As depicted in Fig. 2a–d, the typical framework of ZSM-5 of all samples remains unchanged, regardless with the incorporation of $[Mo_3S_4]^{n+}$ into ZSM-5 zeolite or not, and the results of X-ray diffraction (XRD, Fig. 2e) also confirm the zeolite framework unchanged for all samples. The 10 membered ring of straight cylindrical channel opening to external of ZSM-5 zeolite can be clearly seen in the STEM images, with the pore diameters of about 0.55 nm. Combining Integrated Differential Phase Contrast (iDPC, to image both heavy and light atoms at low irradiation dose) and High-Angle Annular Dark Field (HAADF, with heavier atoms in higher brightness) imaging technology to directly image the zeolite and its pore filler[23], it is confirmed that the ZSM-5 zeolite is indeed filled with molybdenum sulfide clusters, which are surrounded by dotted white circles and pointed by white arrows in enlarged view. The EDX profile obtained from the selected white square also confirms the presence of both S and Mo elements (Supplementary Fig. 2). It can be seen from STEM images that molybdenum sulfide clusters are in the size of around 0.4 nm and do not change or agglomerate before and after the reaction. For the $MoS_x$/NaZSM-5 catalyst, however, it can be clearly seen from the HRTEM image that the lattice fringes correspond to layered $MoS_2$ and the sizes of the $MoS_2$ region are about ~10 nm, much larger than the channel of ZSM-5 zeolite, revealing the $MoS_2$ are supported on the external surface of NaZSM-5 zeolite (Supplementary Fig. 3). As depicted in Fig. 2e, only the diffraction peaks from ZSM-5 zeolite can be seen in the XRD results of $Mo_3S_4$@NaZSM-5 and $MoS_x$/NaZSM-5 catalysts. Even with the $[Mo_3S_4]^{n+}$ clusters embedded into zeolitic pores, the crystallinity of $Mo_3S_4$@NaZSM-5 catalyst does not decay. The typical diffraction patterns related to $MoS_x$ crystals could not be found in all samples (Fig. 2f), implying the highly dispersed state of $MoS_x$. In addition, the diffraction patterns of the $Mo_3S_4$@NaZSM-5 catalyst after 1000 h on stream of $CO_2$ hydrogenation are exactly the same as that of the fresh catalyst, indicative of the $Mo_3S_4$@NaZSM-5 catalyst is extremely stable under reaction conditions.

The coordination statuses of Mo or S in various samples were investigated using X-ray absorption fine structure (XAFS) spectroscopy. According to X-ray absorption near edge structure (XANES) measurements (Fig. 3a), the absorption of Mo K edge in $Mo_3S_4$@NaZSM-5 lacks a peak at a lower energy level (20005 eV), comparing with $MoS_x$/NaZSM-5 and commercial $MoS_2$ (the illustration for details), consistent with the literature results for $[Mo_3S_4]^{n+}$ cluster[24]. The Mo K edge XANES of $MoS_x$/NaZSM-5 is the same with that of commercial $MoS_2$, implying that the $MoS_x$ in $MoS_x$/NaZSM-5 exists in the form of $MoS_2$. The radial distribution functions of various samples obtained by Fourier-transformed extended X-ray absorption fine structure (EXAFS) spectra are shown in Fig. 3b. The coordination numbers and distances of Mo−S shell and Mo−Mo shell are obtained by fitting the radial distribution functions and listed in Table 1. Interestingly, the coordination numbers of Mo−S and Mo−Mo in $Mo_3S_4$@NaZSM-5 are much smaller than that of $MoS_x$/NaZSM-5, but

they are basically close to that of commercial MoS₂ for MoSₓ/NaZSM-5. Combined with the HRTEM image of MoSₓ/NaZSM-5, shown in Supplementary Fig. 3, it can be concluded the external dispersion of MoS₂ on NaZSM-5 zeolite for MoSₓ/NaZSM-5. The size of [Mo₃S₄]ⁿ⁺ clusters in Mo₃S₄@NaZSM-5, about 0.4 nm, is according to the coordination numbers and bond distances obtained through EXAFS measurement, which is also consistent with the results of STEM obervation. Moreover, the bonding of the Mo in [Mo₃S₄]ⁿ⁺ cluster to the framework O of NaZSM-5 zeolite (Fig. 3b)[25] stabilizes the [Mo₃S₄]ⁿ⁺ cluster in the cage of NaZSM-5 zeolite, of which structure is theoretically optimized and shown in Fig. 1d. It is worthy of noting that the absorption of Mo K edge in spent Mo₃S₄@NaZSM-5 shifts slightly to a lower energy in comparison with fresh sample, due to the slight reduction or CHₓ bonding to the Mo₃S₄@NaZSM-5 during hydrogenation reaction, as shown in Fig. 3c. These results are also in accordance with the Mo 3*d* binding energies measured using XPS for the spent Mo₃S₄@NaZSM-5, which move slightly to lower binding energies, compared with the fresh sample (Fig. 3e). Moreoer, it is interesting that both the Mo 3*d* and S 2*p* move slightly to lower binding energies, maybe due to intermediates of hydrocarbons adsorbed on the [Mo₃S₄]ⁿ⁺ cluster during hydrogenation reactions, as presented below. After 1000 h on stream at 180 °C, the coordination states of Mo and S and bond lengths in

Mo₃S₄@NaZSM-5 have not changed and related EXAFS results are presented in Fig. 3d and Table 1, which fully shows that the Mo₃S₄@NaZSM-5 catalyst is extremely stable under typical reaction conditions, though the coordination of oxygen from zeolitic framework to the Mo clusters seems decreased a little after reaction, maybe caused by the tiny movement of the [Mo₃S₄]ⁿ⁺ clusters in channel of ZSM-5 which also results in the binding energy of Mo and S moving to lower positions.

The Mo 3*d* and S 2*p* XPS spectra of related samples are shown in Fig. 3e, f. For both the samples of Mo₃S₄@NaZSM-5 and MoSₓ/NaZSM-5 (Fig. 3e), the binding energies at about 232.6 and 235.8 eV of Mo 3*d*₅/₂ and Mo 3*d*₃/₂ related to MoO₃ cannot be detected, suggesting the complete sulfurization of molybdenum after the Sulphur powder treatment. The Mo 3*d*₅/₂ and Mo 3*d*₃/₂ spectra of MoSₓ/NaZSM-5 are almost the same with that of the commercial MoS₂ and the molybdenum is most likely in state close to Mo⁴⁺ species required by stoichiometry of MoS₂[26], in agreement with the assignment that the molybdenum exists in MoSₓ/NaZSM-5 just as MoS₂ loaded on external surface and weakly interacted with the NaZSM-5. The catalyst Mo₃S₄@NaZSM-5 has much lower ratio of S/Mo, however, its Mo 3*d* binding energy values are higher than those of MoSₓ/NaZSM-5 and commercial MoS₂, for the [Mo₃S₄]ⁿ⁺ clusters

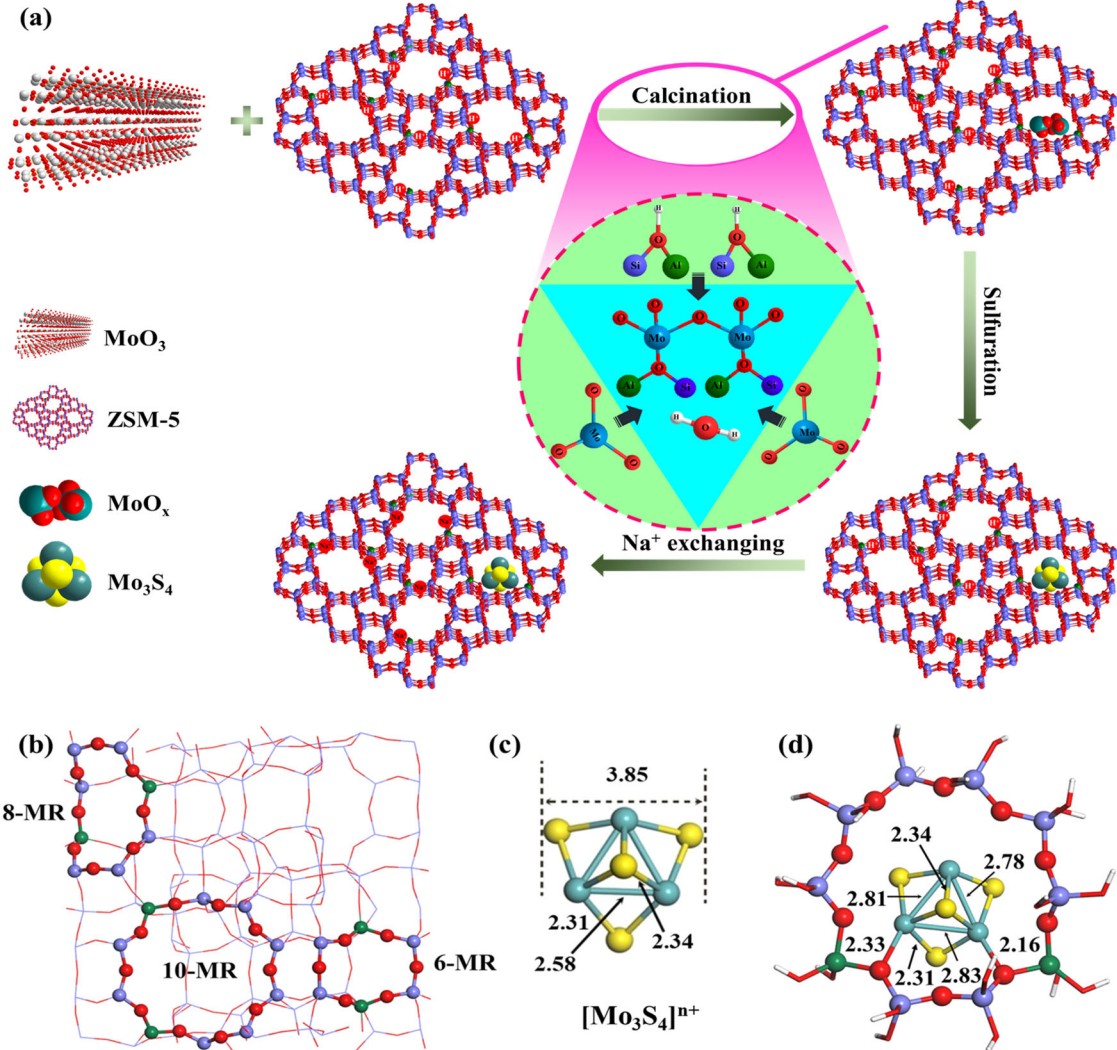

**Fig. 1 | Schematic diagram of catalyst synthesis and structure optimized by DFT calculations. a** Preparation of Mo₃S₄@NaZSM-5 (the shaded region refers to the scheme of solid exchange of MoO₃ with OH groups in HZSM-5, the positions of aluminum and proton are schematically shown.); **b** The zigzag 10-MR(T3/T3), 8- MR(T7/T12), and δ-type 6-MR(T11/T11) Al-pair sites in the ZSM-5 framework; **c** Optimized structure of [Mo₃S₄]ⁿ⁺; **d** The optimized configuration of [Mo₃S₄]ⁿ⁺ intercalated into ZSM-5 molecular sieves containing double aluminum sites in 10-MR. Color legend: Si (purple), Al (green), O (red), Mo (Cyan), and S (yellow).

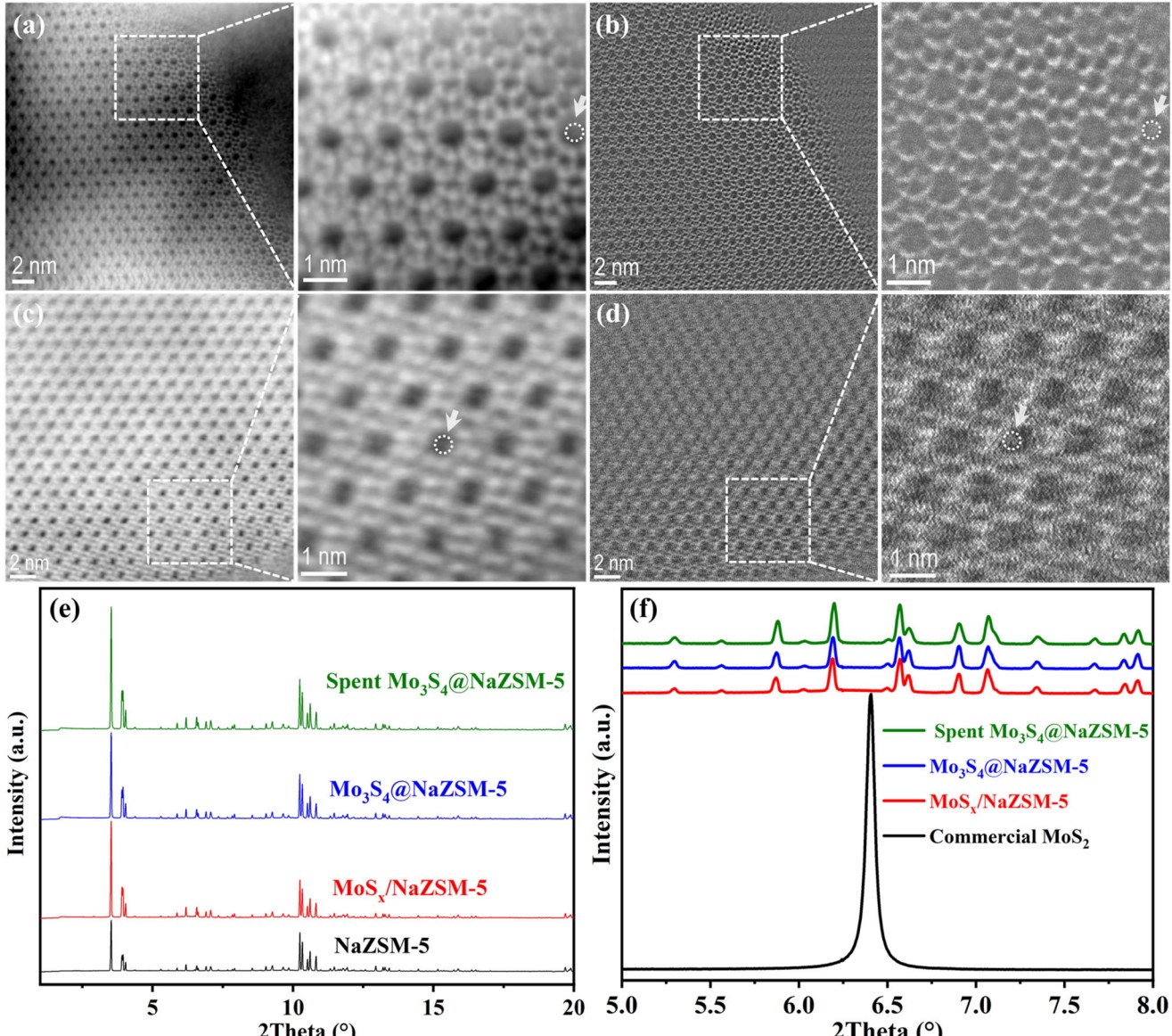

**Fig. 2 | Structural characterizations of catalysts. a** Scanning Transmission Electron Microscope-Integrated Differential Phase Contrast (STEM-iDPC) image of fresh Mo$_3$S$_4$@NaZSM-5 and corresponding enlarged view from the dotted area; **b** Scanning Transmission Electron Microscope-High Angle Annular Dark Field (STEM-HAADF) images of the identical area shown in Fig. 2a and corresponding enlarged view from the dotted area, both STEM-iDPC image and STEM-HAADF image were captured in the same area simultaneously; **c** STEM-iDPC image of the used Mo$_3$S$_4$@NaZSM-5 sample and corresponding enlarged view from the dotted area; **d** STEM-HAADF image of the used Mo$_3$S$_4$@NaZSM-5 sample and corresponding enlarged view from the dotted area; Filled clusters inside the 10 membered ring channels of ZSM-5 zeolite are surrounded by dotted white circles and pointed by white arrows, both STEM-iDPC image and STEM-HAADF image were captured in the same area simultaneously. **e** XRD patterns of NaZSM-5, MoS$_x$/NaZSM-5, Mo$_3$S$_4$@NaZSM-5, and Spent Mo$_3$S$_4$@NaZSM-5; **f** Enlarged view of XRD patterns of commercial MoS$_2$, MoS$_x$/NaZSM-5, Mo$_3$S$_4$@NaZSM-5, and Spent Mo$_3$S$_4$@NaZSM-5 in the range of 5–8°.

in Mo$_3$S$_4$@NaZSM-5 are stabilized in the pores of NaZSM-5 zeolite by the coordination of framework oxygen to Mo, as proved by Fourier-transformed EXAFS spectra of Mo$_3$S$_4$@NaZSM-5 (Fig. 3b). The bonding causes higher binding energy of Mo $3d_{5/2}$ and Mo $3d_{3/2}$ than that in MoS$_x$/NaZSM-5 and commercial MoS$_2$. The binding energy of Mo $3d$ in the spent Mo$_3$S$_4$@NaZSM-5 after 1000 h of reaction moves slightly to lower binding energy, compared with the fresh sample (Fig. 3e), indicating that the slight reduction or CH$_x$ bonding to the Mo$_3$S$_4$@NaZSM-5 during hydrogenation reaction. As depicted in Fig. 3f, the XPS of S $2p$ exhibit trends basically consistent with that of Mo. Interestingly, although the molybdenum species in both catalysts Mo$_3$S$_4$@NaZSM-5 and MoS$_x$/NaZSM-5 are completely sulfurized, the molar ratio of sulfur to molybdenum is very different. As shown in Supplementary Table 1, the molar ratio

of sulfur to molybdenum in MoS$_x$/NaZSM-5 is 1.89, which is close to that of the commercial MoS$_2$ (1.90). However, the S/Mo ratio of Mo$_3$S$_4$@NaZSM-5 is 1.26, close to that of Mo$_3$S$_4$. These results indicate that the status of molybdenum and sulfur in MoS$_x$/NaZSM-5 is similar to MoS$_2$, while the molybdenum and sulfur in Mo$_3$S$_4$@NaZSM-5 are in totally different coordination forms. After 1000 h on stream of reaction, the mole ratio of S/Mo in the spent Mo$_3$S$_4$@NaZSM-5 is still close to 1.26 and no obvious desulfurization is found by XPS measurements.

The Mo contents in samples Mo$_3$S$_4$@NaZSM-5 and MoS$_x$/NaZSM-5 were analyzed by inductively coupled plasma-optical emission spectrometer (ICP-OES) technique and the results are listed in Supplementary Table 2. The 3.01 and 3.03 wt% of Mo contents are obtained respectively for Mo$_3$S$_4$@NaZSM-5 and MoS$_x$/NaZSM-5, close to the

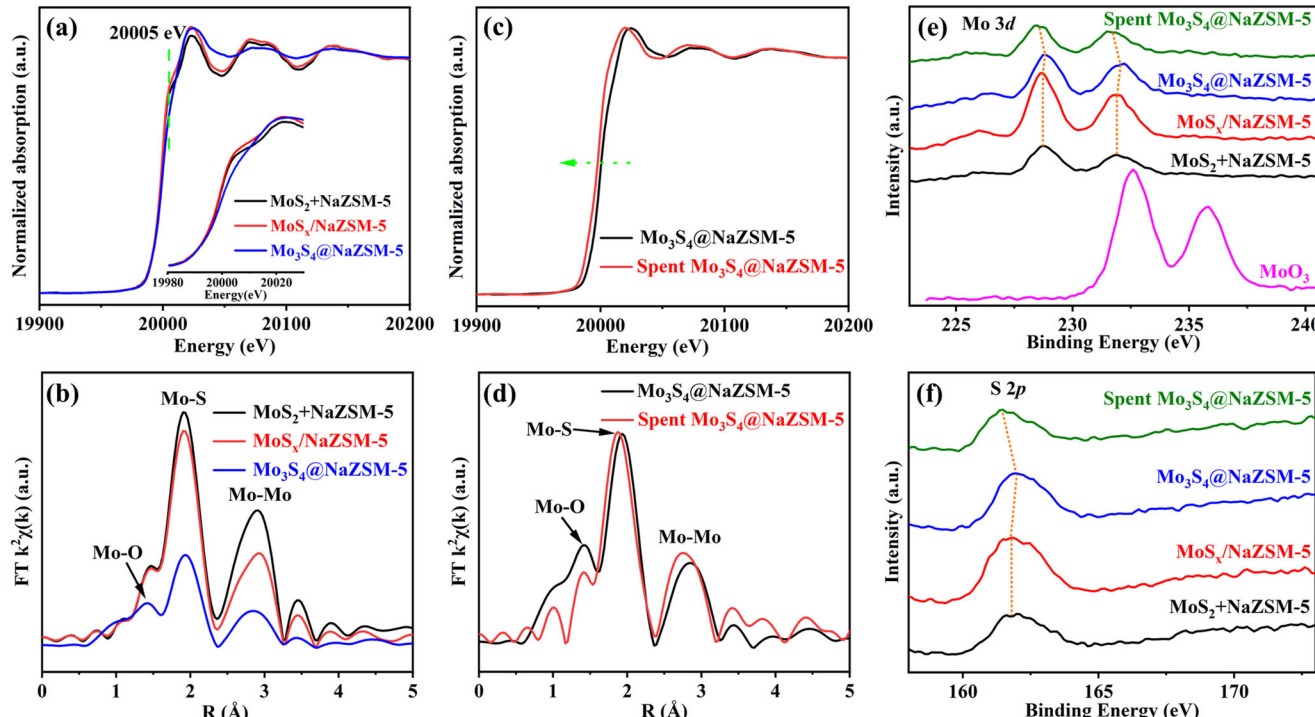

**Fig. 3 | Structure and physical property of catalysts. a** X-ray absorption near edge structure (XANES) spectroscopy and **b** radical distribution functions obtained by Fourier-transformed extended X-ray absorption fine structure (EXAFS) spectra of commercial $MoS_2$, $MoS_x$/NaZSM-5, and $Mo_3S_4$@NaZSM-5; **c** XANES spectroscopy and **d** radical distribution functions obtained by Fourier-transformed EXAFS spectra of $Mo_3S_4$@NaZSM-5 and spent sample; **e** Mo 3$d$ XPS spectra of $MoO_3$, commercial $MoS_2$ + NaZSM-5, $MoS_x$/NaZSM-5, $Mo_3S_4$@NaZSM-5, and Spent $Mo_3S_4$@NaZSM-5; **f** S 2$p$ XPS spectra of commercial $MoS_2$ + NaZSM-5, $MoS_x$/NaZSM-5, $Mo_3S_4$@NaZSM-5, and Spent $Mo_3S_4$@NaZSM-5.

amount of molybdenum added in synthesis. Compared with NaZSM-5 and $MoS_x$/NaZSM-5, $Mo_3S_4$@NaZSM-5 has the lowest Brunauer-Emmett-Teller (BET) specific surface area (Supplementary Table 2), indicating that $[Mo_3S_4]^{n+}$ clusters occupy some zeolitic pores of ZSM-5 zeolite. While little difference is found between the specific surface area of $MoS_x$/NaZSM-5 and NaZSM-5, consistent with the external loading of $MoS_2$ on NaZSM-5 zeolite.

The confinement of metallic centers for hydrogenation by the zeolitic framework in catalyst $Mo_3S_4$@ZSM-5 is further checked by shape-selective hydrogenations of 2,3-dimethylnitrobenzene and nitrobenzene[19,27–31] and the results are listed in Supplementary Fig. 4 and Supplementary Table 3, indicating that $[Mo_3S_4]^{n+}$ is located in the interior of ZSM-5 zeolite for $Mo_3S_4$@ZSM-5, different from that of $MoS_x$/ZSM-5.

### Catalytic hydrogenation of $CO_2$

The catalytic properties of $Mo_3S_4$@NaZSM-5 catalyst for $CO_2$ hydrogenation are measured and depicted in Fig. 4. As shown in Fig. 4a, the

### Table 1 | Curve-fitting results for various samples obtained by Fourier-transformed EXAFS spectra

| Samples | Mo–S | | | Mo–Mo | | |
|---|---|---|---|---|---|---|
| | N[a] | R (Å)[b] | σ²(Å²)[c] | N[a] | R (Å)[b] | σ²(Å²)[c] |
| $Mo_3S_4$@NaZSM-5 | 3.0 ± 0.2 | 2.332 | 0.0025 | 1.9 ± 0.4 | 2.931 | 0.0042 |
| Spent $Mo_3S_4$@NaZSM-5 | 3.0 ± 0.3 | 2.310 | 0.0029 | 2.0 ± 0.3 | 2.93 | 0.0036 |
| $MoS_x$/NaZSM-5 | 5.7 ± 0.3 | 2.321 | 0.0027 | 4.8 ± 0.6 | 3.152 | 0.0045 |
| $MoS_2$ + NaZSM-5 | 6 | 2.321 | 0.0035 | 6 | 3.150 | 0.0039 |

[a]Coordination numbers.
[b]Bond distances.
[c]Debye-Waller factor.

conversion of $CO_2$ increases with the increase of reaction temperature from 120 °C to 270 °C. When the reaction temperature raises from 150 to 180 °C, a conversion jump can be seen, because of the desorption of product methanol from metallic centers is limited by the peripheral NaZSM-5 framework at lower temperatures, as revealed by results of methanol-TPD (Supplementary Fig. 5). At 180 °C, a selectivity to $CH_3OH$ larger than 98% is obtained at $CO_2$ conversion of 10.2% and the selectivity to $CO + CH_4$ is less than 2%. When the reaction temperature exceeds 180 °C, the increase in amount of CO and $CH_4$ lowers the selectivity to $CH_3OH$, driven by thermodynamics (Supplementary Fig. 1). With the pressure rising, as described in Fig. 4b, both the conversion of $CO_2$ and the selectivity of $CH_3OH$ increase, due to the principle of Le Chatelier. Noteworthy, the effect of space velocity on the reaction is very interesting and it can be seen from the Fig. 4c that the $CH_3OH$ selectivity increases and the selectivity to $CO + CH_4$ decreases with the space velocity increasing. It is a little strange but well elucidated by DFT calculations, which clearly show that the energy barrier of *CO desorption is much higher than hydrogenation of *CO to *CHO (see below).

The service life of catalyst is of great importance for industrialization. A fatal problem of the hydrogenation of $CO_2$ reported to date is that the deactivation of the most of supported metal catalysts due to the sintering of active components at high temperatures[32]. For current $Mo_3S_4$@NaZSM-5 catalyst, the active component $[Mo_3S_4]^{n+}$ is protected by the surrounding ZSM-5 zeolitic framework which is extremely stable at relatively low reaction temperature of 180 °C. It keeps catalytic performance, i.e., $CO_2$ conversion >10% and at the same time the $CH_3OH$ selectivity >98%, without any reduction in continuous 1000 h on stream of $CO_2$ hydrogenation (Fig. 4d, 180 °C, 4 MPa, 1200 mL $g_{cat}^{-1}$ h$^{-1}$).

For comparison, the catalysts $MoS_x$/NaZSM-5 and commercial $MoS_2$ are also tested for the reaction and the specific experimental data are listed in Table 2. Interestingly, both of them

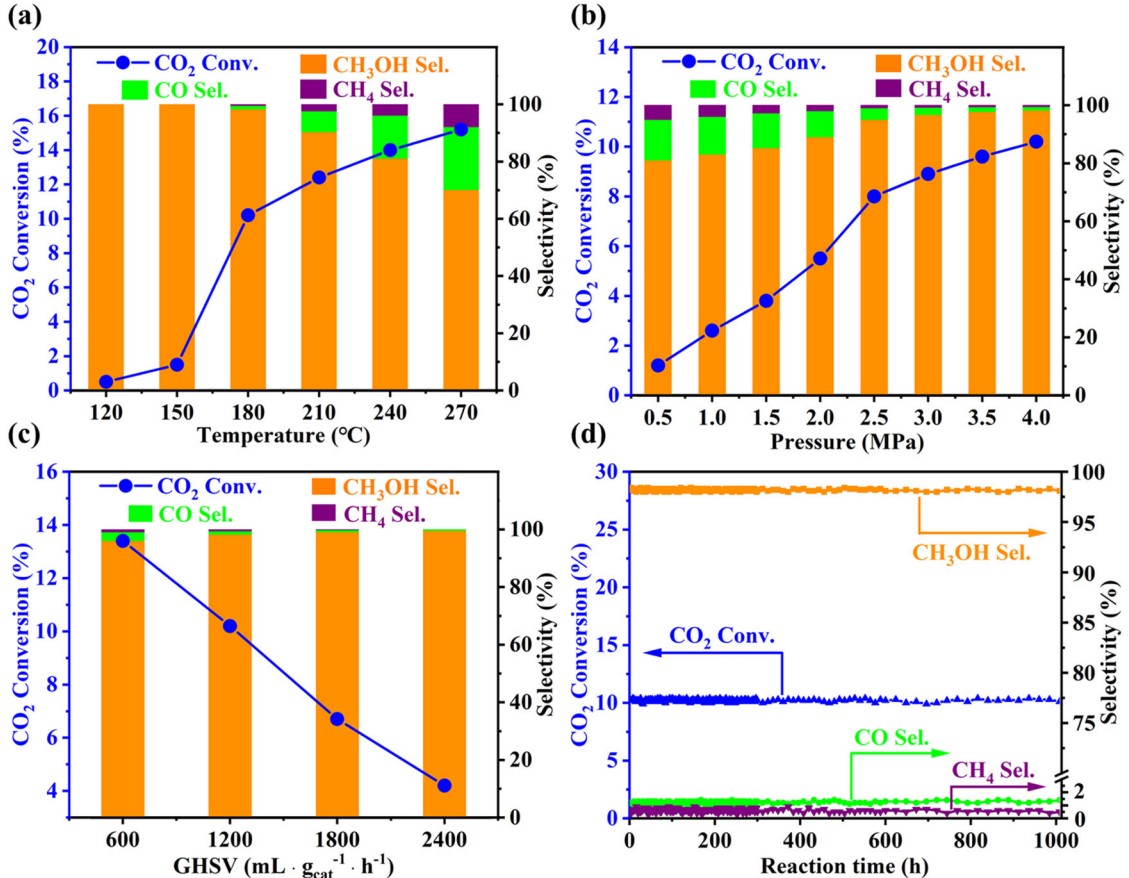

**Fig. 4 | Hydrogenation of CO₂ over Mo₃S₄@NaZSM-5 catalyst with feed gases of 23% CO₂, 69% H₂ and balance Ar. a** Effect of reaction temperature (4 MPa, 1200 mL g$_{cat}^{-1}$ h$^{-1}$); **b** Effect of reaction pressure (180 °C, 1200 mL g$_{cat}^{-1}$ h$^{-1}$); **c** Effect of space velocity (180 °C, 4 MPa); **d** Stability test of the catalyst Mo₃S₄@NaZSM-5 (180 °C, 4 MPa, 1200 mL g$_{cat}^{-1}$ h$^{-1}$). Conv. and Sel. are abbreviations for Conversion and Selectivity, respectively.

exhibit very low catalytic activity for CO₂ hydrogenation under the same reaction conditions. The Mo₃S₄@NaZSM-5 catalyst with active center of [Mo₃S₄]$^{n+}$ and peripheral NaZSM-5 framework surrounding should be a suitable combination showing excellent catalytic performance for CO₂ hydrogenation to CH₃OH. The surrounding NaZSM-5 framework confines and stabilizes the cluster of [Mo₃S₄]$^{n+}$ in zeolitic cages with a small space remained just right for adsorption and reaction of CO₂ and H₂ to methanol, as shown in Fig. 1d.

Over Mo₃S₄@HZSM-5 catalyst, the reaction of CO₂/3H₂ gives 6.4% conversion and 62.5% methanol selectivity (Table 1). Some CO and CH₄ are detected, reflecting the change of electronic properties of [Mo₃S₄]$^{n+}$ clusters modulated by the inner dielectric

field of zeolite upon the replacement of Na⁺ by proton[33]. C₂ hydrocarbons (ethylene and ethane) in selectivity of 24% in the product are obviously from the reaction of CH₃OH at Brønsted acid of HZSM-5.

## Reaction mechanism of CO₂ hydrogenation

Density functional theory (DFT) calculations were carried out to study the reaction mechanism of CO₂ hydrogenation over Mo₃S₄@NaZSM-5 catalyst to understand the relationship between catalyst structure and catalytic activity at the atomic level. The thermodynamic route of CO₂ hydrogenation to methanol and the kinetic energy barriers of key rate-determining steps are shown in Fig. 5a. The schematic diagram of the intermediate structure is shown in Fig. 5b. Firstly, CO₂ is chemisorbed with C and O on Mo that does not bond with 10-MR in Mo₃S₄@NaZSM-5, forming *O-*C-O as shown in Fig. 5b. For this adsorption configuration, the hydrogenation of *CO₂ to form *COOH or *HCOO competes with the direct dissociation of *CO₂ to form *CO and *O formation. By comparing the activation energy barriers, we found that the activation energy barrier for the direct dissociation of *CO₂ to *CO is 0.29 eV, while the activation energy barriers for *CO₂ hydrogenation to *COOH and *HCOO are 0.98 and 0.43 eV respectively. The activation energy barrier for direct dissociation of *CO₂ to *CO is much lower than that for the hydrogenation of *CO₂ to *COOH and *HCOO, which is consistent with our experimental finding that the peaks of *CO was immediately captured in the infrared spectrum when the feeding gas with pure CO₂ was introduced into a chamber in which a self-supported catalyst disk mounted for in-situ infrared measurement without H₂ at room

## Table 2 | Catalytic hydrogenation of CO₂ over related catalysts

| Catalyst | Conversion (%)[a] | Selectivity (%) | | | STY[b] (mg g$_{cat}^{-1}$ h$^{-1}$) |
|---|---|---|---|---|---|
| | | CH₃OH | CH₄/C₂ | CO | |
| MoS₂[c] | 0.3 | 54 | 39/– | 7 | 0.64 |
| MoS$_x$/NaZSM-5 | 0.4 | 63 | 28/– | 9 | 1 |
| Mo₃S₄@NaZSM-5 | 10.2 | 98 | 0.7/– | 1.3 | 39.5 |
| Mo₃S₄@HZSM-5[d] | 6.4 | 62.5 | 5/24 | 8.5 | 15.7 |

Reaction conditions: 180 °C, 4 MPa, 1200 mL g$_{cat}^{-1}$ h$^{-1}$ of GHSV.
[a]Single-pass conversion of CO₂.
[b]Space-time yield of CH₃OH.
[c]Commercial MoS₂ powder, referring to 3 wt% Mo, mixed with NaZSM-5.
[d]Mo₃S₄@NaZSM-5 was prepared from Mo₃S₄@HZSM-5 by neutralization of proton with NaOH.
Calculation methods are given in Methods section.

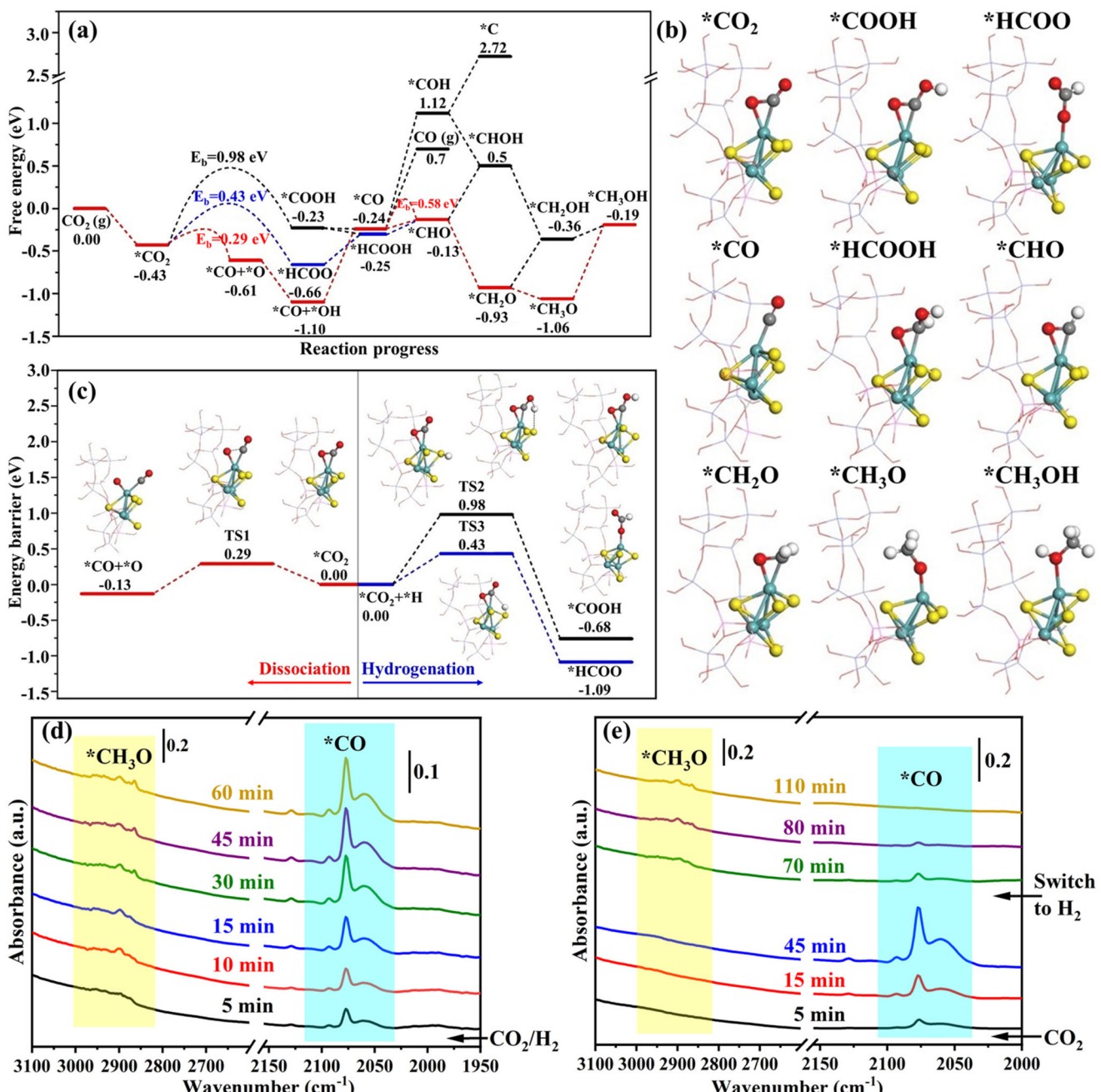

**Fig. 5 | Study on the reaction mechanism of CO₂ hydrogenation to CH₃OH over Mo₃S₄@NaZSM-5.** **a** Free energy diagram of CO₂ hydrogenation to CH₃OH over Mo₃S₄@NaZSM-5 catalyst and kinetic energy barrier (E_b) of the key rate-determining step; **b** Possible intermediates for CO₂ hydrogenation to CH₃OH over Mo₃S₄@NaZSM-5; **c** Potential energy surface of *CO₂ dissociation to *CO and hydrogenation to *COOH and *HCOOH over Mo₃S₄@NaZSM-5 catalyst; **d** Operando FT-IR spectra of Mo₃S₄@NaZSM-5 catalyst during the hydrogenation of CO₂ under the reaction conditions (CO₂/3H₂, 4 MPa, 180 °C); **e** FT-IR spectrums of CO₂ dissociation and after CO₂ was switched to H₂ at room temperature, 4 MPa by introducing feed gas into the chamber with a Mo₃S₄@NaZSM-5 catalyst disk mounted; The structural diagrams of intermediates and transition states (TS) for every step of reaction are shown in the inset. Color legend: Si (purple), Al (green), O (red), Mo (Cyan), S (yellow), and H (white).

temperature (Fig. 5e). The potential energy surface and transition state structure of CO₂ dissociation and hydrogenation are shown in Fig. 5c. The hydrogenation of *CO to *CHO requires remarkably lower activation energies than desorption from the surface of Mo₃S₄@NaZSM-5 catalyst. Importantly, the results also infer that the desorption of *CO, with activation barrier larger than 0.94 eV (0.70 + 0.24), is much difficult than its surface reaction of hydrogenation. The further hydrogenation of *CHO needs lower activation energies to generate CH₃OH through *CH₂O and *CH₃O intermediates. Such reaction paths deduced by DFT calculations

agree well with the results of operando FT-IR characterizations that the appearance of *CO and *CH₃O signals as the feeding gas with CO₂ and H₂ were introduced into the chamber under typical reaction conditions as shown in Fig. 5d. Additionally, when CO₂ was switched to H₂ at room temperature, the peaks of *CO dissociated from CO₂ gradually decreased until disappeared with the increase of *CH₃O signals (Fig. 5e), which further proves that the reaction paths deduced from DFT calculations are completely reasonable. (The IR signals of *CO and *CH₃O species are identified by Ref. [22]).

**Table 3 | Catalytic syngas conversion over Mo₃S₄@NaZSM-5 and Mo₃S₄@HZSM-5**

| Feed gas | Catalyst | Conversion (%)[a] | Selectivity (%) | | |
|---|---|---|---|---|---|
| | | | $CH_3OH$ | $CH_4$ /$C_2$-$C_3$[b] | $CO_2$ |
| $2CO/H_2$ | Mo₃S₄@NaZSM-5 | 2.6 | 97.6 | 2.4/– | 0.8 |
| | Mo₃S₄@HZSM-5 | 3.2 | – | 2/98 | 45 |
| $CO/2H_2$ | Mo₃S₄@NaZSM-5 | 5.4 | 81 | 19/– | 7.6 |
| | Mo₃S₄@HZSM-5 | 5.8 | – | 25/75 | 39 |

Reaction conditions: 260 °C, 4 MPa, 4,000 mL $g_{cat}^{-1}$ $h^{-1}$ of GHSV.
[a]Single-pass conversion of CO.
[b]$C_2$ -$C_3$ hydrocarbons (including olefins and alkanes) selectivity in organic products. Calculation methods are given in Methods section.

## Catalytic conversion of syngas over Mo₃S₄@NaZSM-5 and Mo₃S₄@HZSM-5

We have also tested the catalytic conversion of syngas over catalysts of Mo₃S₄@HZSM-5 and Mo₃S₄@NaZSM-5 and the main results are listed in Table 3. Only $CH_4$ and $CH_3OH$ are detected as organic products over Mo₃S₄@NaZSM-5. For Mo₃S₄@HZSM-5, however, $C_2$ and $C_3$ hydrocarbons are detected in organic products in 98% selectivity, besides 2% $CH_4$, which obviously follows a cascade route, i.e., CO was firstly converted into $CH_3OH$ over [Mo₃S₄]$^{n+}$ clusters and then $CH_3OH$ transformed into light olefins (most olefins are further hydrogenated to alkanes over [Mo₃S₄]$^{n+}$ clusters) at the Brønsted acid of HZSM-5 zeolite.

More details about the reaction of syngas conversion are shown in Fig. 6. Figure 6a shows that the selectivity of $C_2$ + $C_3$

hydrocarbons (olefins and alkanes) in organic products increased from 75% to 98% when the feed gas changed from $CO/2H_2$ to $2CO/H_2$ and the CO conversion simultaneously decreased from 5.8% to 3.2%. It appears that the higher $CO/H_2$ ratio benefits to improve the selectivity to light hydrocarbons. Furthermore, the catalytic performance keeps stable on stream more than 100 h with >98% $C_2$ and $C_3$ hydrocarbons in organic products and CO conversion > 3.2% with $2CO/H_2$ syngas, as shown in Fig. 6b. Figure 6c depicts the results of operando characterization of IR over Mo₃S₄@HZSM-5 during $2CO/H_2$ conversion (260 °C, 4 MPa). The species or intermediates of *$CH_3O$, unsaturated C-H ( =CH), symmetric C = C stretching vibrations, C-C stretching vibrations[34], and methylene bending band (-$CH_2$-)[35] can be observed during reaction. In addition, the absorption peak of $CH_4$ at 3015 $cm^{-1}$ did not appear, indicating that no $CH_4$ formed during the syngas conversion at 260 °C, which is consistent with the results shown in Fig. 6b.

The catalytic stability of Mo₃S₄@HZSM-5 for syngas ($2CO/H_2$) conversion is further tested at 400 °C for 100 h and the results are shown in Fig. 6d. $C_4$ and small amount of $C_{5+}$ hydrocarbons are detected in product. The selectivity of ~90% to $C_2$-$C_4$ hydrocarbons in the organic products is measured at CO conversion of ~20% and the results keep unchanged in 100 h on stream. The results deviate far from the ASF model that predicts the selectivity to $C_2$-$C_4$ hydrocarbons cannot exceed 58%. It is very important that the result of composition analysis on spent catalyst indicate the loss of S is negligible, even on stream of 100 h at 400 °C (Supplementary Table 1). The unique Mo₃S₄@ions-ZSM-5 catalyst, compliant with so-called mesocatalyst in structure we summarized previously[36], would offer opportunities to

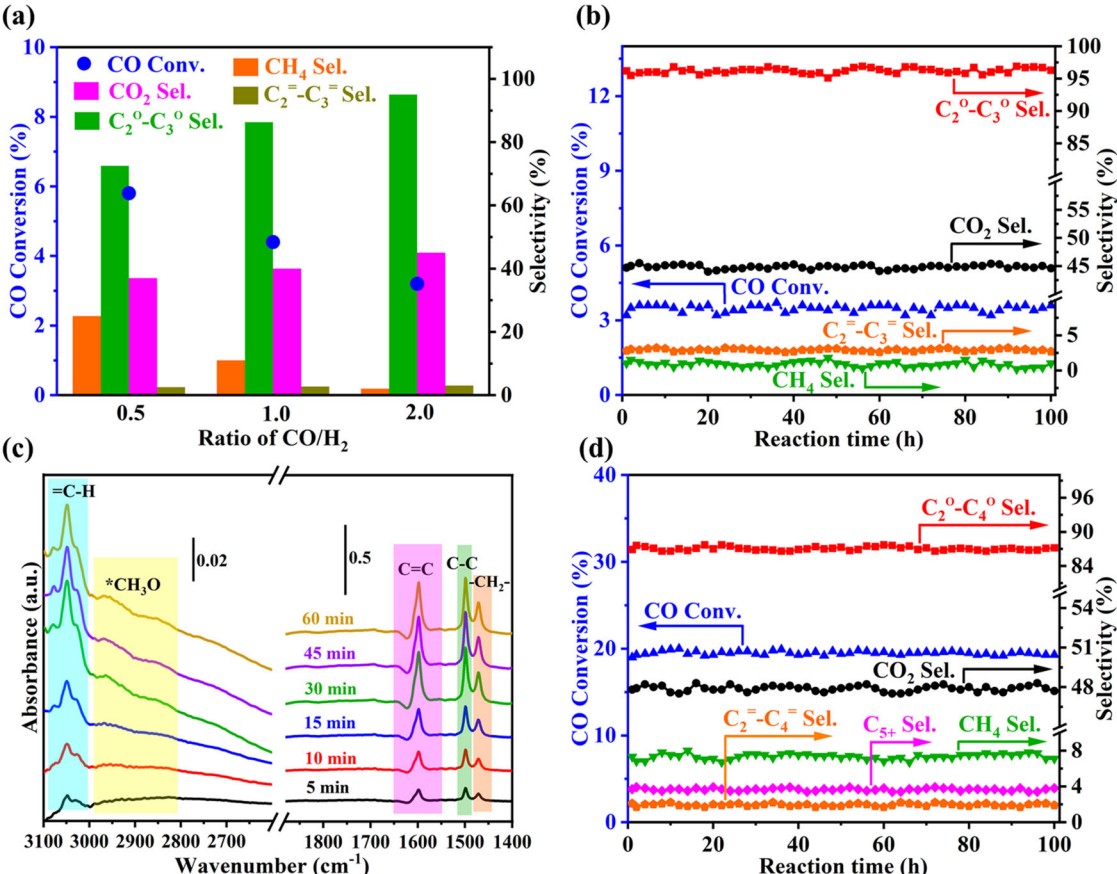

**Fig. 6 | Syngas conversion over Mo₃S₄@HZSM-5. a** Effect of $CO/H_2$ ratio on CO conversion and product distribution (260 °C, 4 MPa, 4,000 mL $g_{cat}^{-1}$ $h^{-1}$); **b** Prolonged test of syngas conversion over Mo₃S₄@HZSM-5 ($2CO/H_2$, 260 °C, 4 MPa, 4,000 mL $g_{cat}^{-1}$ $h^{-1}$); **c** Operando FT-IR spectra of Mo₃S₄@HZSM-5 during the syngas conversion ($2CO/H_2$, 260 °C, 4 MPa); **d** Prolonged test of syngas conversion over Mo₃S₄@HZSM-5 at elevated temperature ($2CO/H_2$, 400 °C, 4 MPa, 4,000 mL $g_{cat}^{-1}$ $h^{-1}$). Conv. and Sel. are abbreviations for Conversion and Selectivity, respectively. = refers to olefins, ° refers to alkanes.

discovery more interesting and important catalytic reactions and further works are under the way.

In summary, the catalyst $Mo_3S_4$@ions-ZSM-5 with $[Mo_3S_4]^{n+}$ clusters embedded in the cages of ZSM-5 zeolite is obtained by two-step solid exchange or reaction method. The properties of the catalyst can be further modified by exchanging the remaining ions with other metallic ions. It is realized the catalyst is capable of dissociating $CO_2$ at room temperature to CO adsorbed on the clusters. With the catalyst $Mo_3S_4$@NaZSM-5, the hydrogenation of $CO_2$ in selectivity more than 98% to methanol and 10.2% conversion of $CO_2$ at 180 °C is confirmed stable at least 1000 h on stream, without any decay. Furthermore, a related catalyst $Mo_3S_4$@HZSM-5 exhibits a peculiar result to catalyze the conversion of syngas with selectivity of $C_2$ and $C_3$ hydrocarbons more than 98% in organics at 260 °C, far beyond the limits of the ASF model. Even at 400 °C for 100 h on stream, the loss of Sulphur from $Mo_3S_4$@HZSM-5 catalyst is negligible. The significance of the catalyst, for conversion of $CO_2$ or syngas, is considered originating from the special combination of the $[Mo_3S_4]^{n+}$ center and the peripheral zeolitic framework. We believe such structural patterns are very effective to design high-performance catalysts.

## Methods

### Catalyst preparation

**$Mo_3S_4$@HZSM-5**. Firstly, 1.5 g HZSM-5 zeolite with the Si/Al molar ratio of 19 and 0.07 g $MoO_3$ (Aladdin Reagent Co., Ltd., 99.9%) were physically mixed and ground in an agate mortar for half an hour. These mixtures were then placed in a tubular furnace and heated at a rate of 5 °C/min in a flowing air to 700 °C and maintained at 700 °C for half an hour. In the step, the $MoO_3$ migrates into the channel of HZSM-5 zeolite and reacts with hydroxyl groups in the zeolite to form $(Mo_2O_5)^{2+}$ dimer and to cling to ZSM-5 framework[18, 20]. Secondly, the $Mo_2O_5$@HZSM-5 was mixed with 1.9 g sulfur powder (Sigma-Aldrich Reagent Co., Ltd., 99.9%) and ground in an agate mortar again for half an hour. Then, the mixture was heated in flowing 10 vol% $H_2/N_2$ at 500 °C for 3 h. The obtained material was denoted as $Mo_3S_4$@HZSM-5 (Mo: 3.01 wt%, obtained from inductively coupled plasma-optical emission spectrometer, ICP-OES).

**$Mo_3S_4$@NaZSM-5**. The $Mo_3S_4$@HZSM-5 was stirred in 0.05 mol/L NaOH ethanol solution (containing 10% water) for one hour at room temperature. The precipitate was filtered out and washed 5 times with ethanol, then dried at 80 °C for 12 h in vacuum. The catalyst is denoted as $Mo_3S_4$@NaZSM-5.

**$MoS_x$/HZSM-5**. By mixing 1.5 g HZSM-5 zeolite, 0.07 g $MoO_3$, and 1.9 g sulfur powder together in an agate mortar and then heating at 500 °C for 3 h in flowing 10 vol% $H_2/N_2$. The Mo content in $MoS_x$/HZSM-5 is 3.03 wt% (obtained by ICP-OES analysis).

**$MoS_x$/NaZSM-5**. $MoS_x$/NaZSM-5 was obtained by neutralization reaction of $MoS_x$/HZSM-5 in 0.05 mol/L NaOH ethanol/water (9:1) solution.

### Hydrogenation of 2,3-dimethylnitrobenzene and nitrobenzene

Typically, 0.3 g catalyst $Mo_3S_4$@HZSM-5 or $MoS_x$/HZSM-5, 1 mmol 2,3-dimethylnitrobenzene, 1 mmol nitrobenzene and 20 mL ethanol were added to a 50 mL Teflon-lined autoclave. The autoclave was then filled with 2 MPa of $H_2$, after the replacement of air in the reactor for five times by hydrogen. Finally, the autoclave was heated at 100 °C for a period of time with a magnetic stirring rate of 400 r/min. The conversion of reactants and selectivity to products were determined by gas chromatography using p-xylene as internal standard.

### Hydrogenation of $CO_2$ in fixed bed reactor

Catalytic hydrogenation of $CO_2$ was performed in a tubular continuous-flow, fixed-bed reactor

equipped with a gas chromatograph (GC-9860). In a typical procedure, the catalyst was pressed and crushed to particles between 20 and 40 mesh. 0.5 g catalyst was then placed in a U-shaped reaction tube (316 L stainless steel) with an inner diameter of ~4 mm. A mixed gas (23% $CO_2$, 69% $H_2$ and balance Ar) was used to measure the catalytic property for hydrogenation of $CO_2$. Products were analyzed using an on-line gas chromatography equipped with a FID detector and two thermal conductivity detectors (TCD). The Ar in feed gases was used as internal standard.

The conversion and selectivity were calculated as follows:

$$CO_2 \text{ Conversion} = \frac{CO_{2\,inlet} - CO_{2\,outlet}}{CO_{2\,inlet}} \times 100\% \quad (5)$$

$$CH_3OH \text{ Selectivity} = \frac{CH_3OH_{outlet}}{CO_{outlet} + CH_{4\,outlet} + CH_3OH_{outlet}} \times 100\% \quad (6)$$

$$CO \text{ Selectivity} = \frac{CO_{outlet}}{CO_{outlet} + CH_{4\,outlet} + CH_3OH_{outlet}} \times 100\% \quad (7)$$

$$CH_4 \text{ Selectivity} = \frac{CH_{4\,outlet}}{CO_{outlet} + CH_{4\,outlet} + CH_3OH_{outlet}} \times 100\% \quad (8)$$

Where $CO_{2\,inlet}$ and $CO_{2\,outlet}$ respectively represent the amount of $CO_2$ entering and flowing out of the reactor, referenced to inner standard argon. The $CH_3OH_{outlet}$, $CO_{outlet}$ and $CH_{4\,outlet}$ denote the amount of $CH_3OH$, CO and $CH_4$ in product, referenced to inner standard argon.

For syngas conversion:

$$CO \text{ Conversion} = \frac{CO_{inlet} - CO_{outlet}}{CO_{inlet}} \times 100\% \quad (9)$$

$$CO_2 \text{ Selectivity} = \frac{CO_{2\,outlet}}{CO_{inlet} - CO_{outlet}} \times 100\% \quad (10)$$

$$C_aH_b \text{ Selectivity} = \frac{aC_aH_{b\,outlet}}{\sum_1^a aC_aH_{b\,outlet}} \times 100\% \quad (11)$$

Where $C_aH_b$ Selectivity denotes the individual hydrocarbon selectivity in organic products. The carbon balance before and after the reaction exceeds 95%.

### Characterization techniques

ICP-OES Avio500 instrument produced by PerkinElmer Co., Ltd was used to analyze the composition of catalysts. Micromeritics ASAP 2010 analyzer was used to measure the specific surface area of samples at liquid nitrogen temperature. X-ray diffraction (XRD) data were collected at BL14B1 beamline of Shanghai Synchrotron Radiation Facility from 0.8° to 25°. X-ray photoelectron spectroscopy (XPS) and Electron Spectroscopy for Chemical Analysis (ESCA) measurements were performed on a PHI5000 Versa Probe XPS system equipped with Al Kα X-ray as exciting source, the results were measured by transferring the samples to the test chamber under vacuum. The XAS spectra were collected at BL14W1 beamline of Shanghai Synchrotron Radiation Facility, for analysis of XANES and EXAFS. HAADF-STEM characterization was performed on a Thermo Fisher Themis Z transmission electron microscope (operated at 300 kV). The STEM was equipped with a probe corrector, monochromator, HAADF detector, segmented DF4 detector and SuperX EDS system. This instrument enables us to obtain

a spatial resolution of 60 pm under STEM mode. The high-resolution iDPC-STEM (acquired by four-quadrant segmented detectors) images and corresponding STEM-HADDF images were acquired simultaneously with a convergence semi-angle of 25 mrad. The HAADF and iDPC-STEM images were acquired at collection angles of 21–127 and 5–20 mrad, respectively. The range of beam current was 0.5–1.5 pA and no beam damage on zeolite was observed during the STEM characterization. The elemental mappings were acquired by SuperX EDS system.

$CH_3OH$ temperature-programmed desorption ($CH_3OH$-TPD) was carried out on a TP5076 apparatus (Tianjin Xianquan industry and Trade Development Co., Ltd) equipped with a TCD. Operando Fourier transform infrared spectroscopy (FT-IR) techniques consisting of a Bruker TENSOR 27 spectrometer and an in-situ IR cell with $CaF_2$ windows which can be heated and pressurized. Typically, ~10 mg of $Mo_3S_4$@NaZSM-5/$Mo_3S_4$@HZSM-5 catalyst was pressed to disk and placed in the IR cell and pretreated in high purity helium of 30 mL/min at 180/260 °C for one hour. Then the helium was switched to $CO_2/3H_2$ or $2CO/H_2$ and the pressure was raised to 4 MPa. Finally, the infrared spectra were recorded with interval of 5-min to 110 min.

## Computational methods

All spin polarized DFT calculations were performed with the Vienna Ab initio simulation package (VASP)[37–39]. The exchange correlation function was handled using the generalized gradient approximation (GGA) formulated by the Perdew-Burke-Ernzerhof (PBE)[40, 41]. The plane-wave basis set energy cutoff was set to 400 eV, and the Brillouin zone integrations were done using a $1 \times 1 \times 1$ k-point mesh. The electronic self-consistency criterion was set to $10^{-4}$ eV and a force convergence tolerance of 0.02 eV/Å.

To investigate $CO_2$ reduction mechanism, we have taken the climbing-image nudged elastic band method (CI-NEB), and the nature of transition state was verified by only one imaginary frequency existing in vibrational normal mode. The convergence criteria were $1 \times 10^{-4}$ eV energy differences for solving for the electronic wave function for transition states (TS), and a force convergence tolerance of 0.02 eV/Å.

The Gibbs free energy $\Delta G$ of $CO_2$ to form $CH_3OH$ is defined as $\Delta G = \Delta E + \Delta E_{ZPE} - T\Delta S$. There, $\Delta E$, $\Delta E_{ZPE}$, and $\Delta S$ are the energy change, the zero-point energy (ZPE) change and entropy change of the reaction, respectively. The zero-point energy and entropy of all intermediates and gaseous molecules are obtained by using the VASPKIT code[42].

## Computational models

The ZSM-5 zeolite model was constructed using a periodic MFI unit cell with an experimental lattice constant of 19.88 Å × 20.11 Å × 13.37 Å. Some of Si atoms were substituted by Al atoms, resulting in a Si/Al ratio of 19. As shown in Fig. 1b, we consider three distinct ring sites of the MFI zeolite framework, namely the zigzag ten-membered ring (10-MR), 8-MR, and 6-MR sites, for hosting $[Mo_3S_4]^{n+}$ in Fig. 1c. Due to the ring sizes, $[Mo_3S_4]^{n+}$ is hosted only on the large 10-MR site. For 10-MR, we take a rigid frame structure and O is saturated with H, as shown in Fig. 1d. All -OH are fixed while the rest of the atoms are released.

As shown in Fig. 1d, the Mo-Mo bond length will be elongated and the Mo-S bond length remains basically unchanged when $[Mo_3S_4]^{n+}$ is intercalated into 10-MR, $[Mo_3S_4]^{n+}$ forms chemical bonds with two oxygen atoms in 10-MR, and the bond lengths are 2.33 and 2.16 Å respectively. And these two oxygen atoms are the meta positions of 10-MR Oxygen, respectively, forms chemical bonds with Al.

## Data availability

Source data are provided with this paper.

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

## Acknowledgements
This work was supported by the National Natural Science Foundation of China (21932004, 91963206, 21872067, 22172072, and 22072090) and the Ministry of Science and Technology of China (2021YFA1500301). The support from the NJU-HUACHANG Joint Institute of Meso Catalysis was also appreciated.

## Author contributions
W.D. conceived the work. G.L. desined and performed main experiments, and D.M., T.Z., X.Q., and Q.H. joint the experimental work. P.F.L. and Q.W. were in charge of computational calculations. J.Q., L.C., and X.L. were in charge of HRTEM characterization. J.M. organized XAS experiments and analyzed the data. X.G., L.M.P., N.X., and Y.Z. joint the experiment and data discusssions. G.L. and W.D. wrote the manuscript and W.D. finalized the work. All authors discussed the results.

## Competing interests
The authors declare no competing interests.
