## [Peer Review File · Nature Communications]

CO_x Hydrogenation to Methanol and Other Hydrocarbons under Mild Conditions with Mo₃S₄@ZSM-5REVIEWER COMMENTS

Reviewer #1 (Remarks to the Author):

This manuscript reported the molybdenum sulfide cluster confined in channel of ZSM-5 as a catalyst for the hydrogenation of CO_x to methanol and hydrocarbons. The Mo₃S₄@NaZSM-5 catalyst exhibited good activity and selectivity in CO₂ hydrogenation to methanol. More interesting, the Mo₃S₄@HZSM-5 catalyst presented high selectivity to C₂-C₃ hydrocarbons when it was used for CO hydrogenation, demonstrating the potential of molybdenum sulfide-based composite catalysts in CO hydrogenation to C₂+ products under mild reaction conditions. Therefore, I think the background of CO_x hydrogenation to higher hydrocarbons should also be introduced in the section of "Introduction". Overall, this work is an interesting contribution that broadens our materials basis in CO_x hydrogenation, especially for the synthesis of C₂+ products. I agree the publication of this work after the following concerns being solved.

1. In the section of "Introduction", the authors discussed the thermodynamic calculation on the reaction of CO₂ hydrogenation. It should be noted that CO is not the only by-product in methanol synthesis from CO₂ hydrogenation. If the reaction of CO₂ hydrogenation to hydrocarbons was also considered in such a thermodynamic calculation, the equilibrium composition would be different? Actually, it is unclear that whether the thermodynamic calculation is suitable for estimating the equilibrium product selectivity, especially in such a complex reaction network.
2. The authors said that "EDX profile obtained from the selected yellow square also confirms presence of both S and Mo." Correspondingly, the EDX data should be provided.
3. The Method detail of ESCAR in "Supporting Information" should be provided.
4. As for the characterization of Mo_xS_y/NaZSM-5 catalyst, it would be better if the authors can provide TEM results as direct evidence for demonstrating the MoS₂ is supported on the external surface of NaZSM-5. The similar XANES and XPS spectra between Mo_xS_y/NaZSM-5 and commercial MoS₂ can only give speculative conclusion.
5. What caused the reduction of Mo in Mo₃S₄@NaZSM-5 after the CO₂ hydrogenation reaction? It seems that the Mo-O coordination number decreased after the reaction. Is that the reduction of the catalyst was caused by the loss of surface oxygen during the reaction? The authors should provide further evidence or discussion correspondingly.
6. The activation energy for the desorption of CO from the surface of Mo₃S₄@NaZSM-5 should be provided to support the conclusion that "the hydrogenation of *CO to *CHO requires remarkably lower activation energies than desorption from the surface of Mo₃S₄@NaZSM-5 catalyst".
7. The composition analysis data should be provided to support the conclusion that "the loss of S is negligible even on stream 100 h at 400 oC."

Reviewer #2 (Remarks to the Author):

In this work, the Mo₃S₄@ions-ZSM-5 catalyst was prepared for CO₂ hydrogenation and syngas

conversion. However, the structure of Mo₃S₄@ions-ZSM-5 was not revealed clearly and little new insights on catalysis was provided in this work. Thus, this reviewer think that the manuscript is quite insufficient for publication in Nature Communications.

- (1) Figure 1a: the schematic diagram of OH group in HZSM-5 was wrong.
- (2) Line 120: it is hard to observe that the Mo_x cluster was embedded into the pores of zeolite.
- (3) Line 121: the EDX profile was not provided.
- (4) Line 144: this can not confirm the external dispersion of MoS₂ on NaZSM-5 zeolite in Mo_x/NaZSM-5.
- (5) Line 145: How is the size of [Mo₃S₄]_n⁺ clusters in Mo₃S₄@NaZSM-5 calculated?
- (6) Figure 2k-2l: the shift of binding energy in XPS spectra is generally related to the electron transfer from one element to another. Why do S and Mo both move to lower binding energy after reaction?
- (7) Figure 2k-2l: XPS is a surface analysis method. If the Mo_x cluster was embedded into the pores of zeolite, it should be very hard to detect the S and Mo elements. Why is the intensity of S and Mo peaks in Mo₃S₄@NaZSM-5 so strong?
- (8) Line 176: XPS spectrum of MoO₃ should be added.
- (9) Table S1: How is the S/Mo ratio measured? The ESCAR characterization method is not explained in the section of Characterization techniques.
- (10) The calculation formula for CH₃OH selectivity was unreasonable.
- (11) Table 3: there seems to be something wrong with the data of product selectivity. The O element during reaction is imbalance.
- (12) Line 321: why did the hydrocarbons distribution over Mo₃S₄@HZSM-5 deviate far from the ASF model?

Reviewer #3 (Remarks to the Author):

The manuscript “Mo₃S₄@ZSM-5 for Highly Efficient Hydrogenation of CO_x to Methanol and Hydrocarbons under Mild Reaction Conditions”, submitted by Liu et al, is dedicated to the important research on the direct conversion of CO₂ and CO into valuable chemicals and contains quite a number of interesting results. Through appropriate modification of the Mo₃S₄@ZSM-5 catalyst, both CO₂ and CO can be converted into methanol and low hydrocarbons in high selectivity. Various important characterization means are explained and presented in good manners. This reviewer recommends the acceptance of the work for publication by Nature Communications after minor revisions, considering the following points.

1. Keywords. Mo₃S₄ cluster should be written as [Mo₃S₄]_n⁺ cluster.
2. Why the ZSM-5 zeolite was used in this work and have the authors explored the influence of Si/Al on the catalyst? Whether other types of zeolites are feasible?
3. Catalyst preparation. Can NaZSM-5 zeolite be directly used to synthesize Mo₃S₄@NaZSM-5 by solid exchange method?
4. Figure S2. The cross-sectional diameter of 2,3-dimethylnitrobenzene is slightly larger than the channel diameters of ZSM-5, but the Mo₃S₄@HZSM-5 catalyst still shows certain catalytic activity during the hydrogenation of 2,3-dimethylnitrobenzene. Is this caused by part of Mo_x not entering the micropores

of ZSM-5 zeolite?

5. Why do Mo₃S₄@NaZSM-5 and Mo₃S₄@HZSM-5 show excellent performance in CO₂ and CO hydrogenation reactions, respectively? How alkali metals affect the electronic structure of molybdenum species? What is the nature of the active site?

6. As far as I know, the ZSM-5 is often used to synthesize C₅+ hydrocarbons in syngas conversion. However, the selectivity of C₂-C₃ hydrocarbons is more than 98% in organics on Mo₃S₄@HZSM-5. Why?

Re : NCOMMS-22-24235-T

Response to Reviewers

Reviewer #1 (Remarks to the Author): This manuscript reported the molybdenum sulfide cluster confined in channel of ZSM-5 as a catalyst for the hydrogenation of CO_x to methanol and hydrocarbons. The $\text{Mo}_3\text{S}_4@NaZSM-5$ catalyst exhibited good activity and selectivity in CO_2 hydrogenation to methanol. More interesting, the $\text{Mo}_3\text{S}_4@HZSM-5$ catalyst presented high selectivity to $\text{C}_2\text{-C}_3$ hydrocarbons when it was used for CO hydrogenation, demonstrating the potential of molybdenum sulfide-based composite catalysts in CO hydrogenation to C_{2+} products under mild reaction conditions. Therefore, I think the background of CO_x hydrogenation to higher hydrocarbons should also be introduced in the section of “Introduction”. Overall, this work is an interesting contribution that broadens our materials basis in CO_x hydrogenation, especially for the synthesis of C_{2+} products. I agree the publication of this work after the following concerns being solved.

Response: Thanks for the good suggestion. The section of “Introduction” has been modified, including the background about catalysts used in CO_2 or CO hydrogenation to hydrocarbons. Corresponding changes were marked in the revised manuscript. (lines 29-34, pages 2 and line 1, page 3)

1. In the section of “Introduction”, the authors discussed the thermodynamic calculation on the reaction of CO_2 hydrogenation. It should be noted that CO is not the only by-product in methanol synthesis from CO_2 hydrogenation. If the reaction of CO_2 hydrogenation to hydrocarbons was also considered in such a thermodynamic calculation, the equilibrium composition would be different? Actually, it is unclear that whether the thermodynamic calculation is suitable for estimating the equilibrium product selectivity, especially in such a complex reaction network.

Response: It is a good question. In current work with the catalyst of $\text{Mo}_3\text{S}_4@NaZSM-5$, only methanol, CO and extremely small amount of CH_4 are experimentally found as products. We have thus calculated the species distribution at thermodynamic equilibrium among CO_2 , CO, H_2 , and CH_3OH , as an understanding of reaction tendency, though the true mixed products at the outlet of

reactor may be far deviated from the equilibrium. If hydrocarbons are included in calculation, the situation becomes complicated, and the thermodynamic tendency of the reaction is similar. In revision, methane, as representative of hydrocarbons, was added to the reaction system for thermodynamic calculation and the results are included in Figure S1. As shown in the Figure, methane has thermodynamic advantages over the CO or methanol as reaction product of CO₂ hydrogenation. This means the current catalyst Mo₃S₄@NaZSM-5 is very active for the cleavage of the first C-O bond of carbon dioxide, in agreement with IR results, and almost inactive for activation of carbon monoxide, considering the high selectivity to methanol experimentally detected. (lines 23-28, page 2 and Supporting Information)

Figure S1. The equilibrium values for hydrogenation of CO₂ to (a) CH₃OH, CO and CH₄ and (b) CH₃OH, CO in the reaction temperature range from 150 °C to 300 °C (Selectivity to water is not included).

2. The authors said that “EDX profile obtained from the selected yellow square also confirms presence of both S and Mo.” Correspondingly, the EDX data should be provided.

Response: Yes, according to the suggestion, the EDX data are added in Figure S2 and corresponding changes are highlighted in the revised manuscript. (lines 14-15, page 5 and Supporting Information)

Figure S2. The EDX profile of (a) fresh and (b) spent (1000 h) Mo₃S₄@NaZSM-5, analyzed at the selected yellow square in Figure 2 (a-d).

3. The Method detail of ESCAR in “Supporting Information” should be provided.

Response: The acronym “ESCA” means “Electron Spectroscopy for Chemical Analysis” was wrong written as “ESCAR” in the original manuscript. Thanks very much for reminding. The explanation was added to the Table S1 in revision (Supporting Information). And the method detail is added in Characterization techniques in revision and related sentences of revised manuscript are highlighted (lines 15-18, page 17). Thanks again for your very careful reading.

4. As for the characterization of MoS_x/NaZSM-5 catalyst, it would be better if the authors can provide TEM results as direct evidence for demonstrating the MoS₂ is supported on the external surface of NaZSM-5. The similar XANES and XPS spectra between MoS_x/NaZSM-5 and commercial MoS₂ can only give speculative conclusion.

Response: It is a very good suggestion and we have added the HRTEM image of MoS_x/NaZSM-5 as Figure S3 (Supporting Information), in which the lattice fringes correspond to layered MoS₂ and

the sizes of the MoS₂ region are about ~10 nm, much larger than the channel of ZSM-5 zeolite.

Corresponding revision has been made. (lines 16-20, page 5 and lines 4-6, page 6)

Figure S3. The HRTEM image of MoS_x/NaZSM-5 catalyst. The lattice fringes correspond to layered MoS₂ and the sizes of the MoS₂ region are about ~10 nm, much larger than the channel of ZSM-5 zeolite.

5. What caused the reduction of Mo in Mo₃S₄@NaZSM-5 after the CO₂ hydrogenation reaction? It seems that the Mo-O coordination number decreased after the reaction. Is that the reduction of the catalyst was caused by the loss of surface oxygen during the reaction? The authors should provide further evidence or discussion correspondingly.

Response: Thanks for the careful reading. In fact, the catalyst Mo₃S₄@NaZSM-5 shows an extreme stability during 1000 h CO₂ hydrogenation reaction both in chemical composition and catalytic property. The reduction of Mo in Mo₃S₄@NaZSM-5 after the CO₂ hydrogenation reaction, as disclosed by Mo 3d binding energy (XPS, Figure 2k), should be caused by some fragments of hydrocarbon intermediates (revealed by IR measurements) adsorbed in the catalyst and on the Mo₃S₄ clusters, which cause the catalyst in more reduced state. As to the decrease in oxygen coordination of the first shell from the zeolitic oxygen (Figure 2j), it would be caused by the tiny movement of the Mo₃S₄ in the zeolitic channels, due to the adsorption of hydrocarbon intermediates. (lines 15-17 and 20-23, page 6)

6. The activation energy for the desorption of CO from the surface of Mo₃S₄@NaZSM-5 should be provided to support the conclusion that “the hydrogenation of *CO to *CHO requires remarkably lower activation energies than desorption from the surface of Mo₃S₄@NaZSM-5 catalyst”.

Response: Thanks for the thoughtful question. The Gibbs free energy of *CO dissociation from the Mo₃S₄@NaZSM-5 surface and the activation energy barrier of *CO hydrogenation to *CHO were further considered in revision. (Figure 4 (a), page 13 and lines 7-10, page 12)

In general, the activation energy of a reaction must be greater than 0 and also must be greater than the reaction energy. In theoretical calculations, the activation energy barriers for the dissociation and adsorption of a gas molecule are difficult to accurately describe. As shown in Figure 4 (a), the Gibbs free energy of CO dissociation from the Mo₃S₄@NaZSM-5 catalyst surface is 0.94 eV, which is a strong endothermic process (entropy change compensation is far from enough to offset the enthalpy change). Therefore, the activation energy barrier for *CO dissociation from the Mo₃S₄@NaZSM-5 surface must be greater than 0.94 eV. Further calculation results show that on the surface of Mo₃S₄@NaZSM-5, the activation energy barrier of *CO hydrogenation to *CHO is 0.58 eV, which is much lower than 0.94 eV, as shown in Figure 4 (a). Therefore, it is reasonable to explain “the hydrogenation of *CO to *CHO requires remarkably lower activation energies than desorption from the surface of Mo₃S₄@NaZSM-5 catalyst”.

We have provided the energy for the desorption of *CO from the surface of Mo₃S₄@NaZSM-5 in Figure 4 (a) in revised manuscript.

Figure 4 (a). Free energy diagram of CO₂ hydrogenation to CH₃OH over Mo₃S₄@NaZSM-5 catalyst and kinetic energy barrier (E_b) of the key rate-determining step.

7. The composition analysis data should be provided to support the conclusion that “the loss of S is

negligible even on stream 100 h at 400 °C.”

Response: The composition analysis data of spent Mo₃S₄@HZSM-5 at 400 °C is added in Table S1.

Thanks a lot. (Supporting Information)

Reviewer #2 (Remarks to the Author): In this work, the Mo₃S₄@ions-ZSM-5 catalyst was prepared for CO₂ hydrogenation and syngas conversion. However, the structure of Mo₃S₄@ions-ZSM-5 was not revealed clearly and little new insights on catalysis was provided in this work. Thus, this reviewer think that the manuscript is quite insufficient for publication in Nature Communications.

Response: Thanks for the forthright remarks. As Prof. Iglesia’s work, the structure and catalytic property of Mo₂O₅@HZSM-5 and MoC_x@HZSM-5 for methane conversion have been reported at 2001 (*J. Phys. Chem. B* 2001, 105, 506) and MoN_x@HZSM-5 was also reported for ammonia synthesis at 2010 (*ChemCatChem* 2010, 2, 167). The catalysts Mo₃S₄@ions-ZSM-5, however, have not been investigated before current work. The catalytic results for hydrogenation of CO₂ to methanol over Mo₃S₄@ions-ZSM-5 also indicate the investigation on Mo₃S₄@ions-ZSM-5 is valuable. The characterizations to the synthesis and structure of Mo₃S₄@ions-ZSM-5 catalysts are carried out as far as possible by all modern research techniques, including XRD, XPS, in-situ IR, XAS (XANES and EXAFS), and HAADF-STEM operated at 300 kV. DFT calculations are also performed to analyze the structure of Mo₃S₄@ions-ZSM-5 and the results are well in agreement with experimental. The structure of Mo₃S₄@ZSM-5 catalyst, in configuration of [Mo₃S₄]ⁿ⁺ incorporated in ZSM-5 cages containing double aluminum sites in 10-MR, is reasonable and also supported by all experimental results. Through appropriate modification of the Mo₃S₄@ions-ZSM-5 catalysts, high-performance catalytic hydrogenation of CO₂ to CH₃OH (Mo₃S₄@Na-ZSM-5) and catalytic hydrogenation of CO to low hydrocarbons (Mo₃S₄@H-ZSM-5) under mild reaction conditions. These investigations are systematic, very important and never been reported in literatures before. We highly appreciate the comments from the reviewer, who spent many times to read our manuscript and asked many thoughtful questions which are remarkably helpful to modify the research work. Thanks very much again.

1. Figure 1a: the schematic diagram of OH group in HZSM-5 was wrong.

Response: Thanks for the comment. Indeed, we show protons in the HZSM-5 channels in Figure

1a just as a diagram. It is not precise position or bonding of proton to the framework of zeolite. For more clarity, it has been modified in revision and hope it is acceptable now. Thanks again.

2. Line 120: it is hard to observe that the MoS_x cluster was embedded into the pores of zeolite.

Response: Thanks a lot. As a great progress of electron microscope, the HAADF and iDPC techniques can image the filler in the micropores of ZSM-5 zeolite (see *Nat. Catal.* 2020, 3, 628) and determine subnanometric clusters confined in the 10R channels of ZSM-5 zeolite. Prof. Xi Liu, one of authors of current manuscript, is a high-level expert in the field. The figures are further processed for more clarity in revision. And, the EDX data obtained at the yellow square of Figure 2 are added as Figure S2 in revision, which show the existence of Mo and S.

In addition, the insertion of MoS_x cluster into the micropores of ZSM-5 zeolite was further verified by the hydrogenations of 2,3-dimethylnitrobenzene and nitrobenzene, which is commonly used to detect the confinement or not by the zeolite framework to the clusters (see *J. Am. Chem. Soc.* 2012, 134, 17688-17695; *ACS Catal.* 2015, 5, 6893; *Angew. Chem. Int. Edit.* 2016, 128, 9324; *Nat. Commun.* 2017, 8, 15240; *Nat. Mater.* 2017, 16, 132; *Nat. Chem.* 2012, 4, 1030; *J. Am. Chem. Soc.* 2015, 137, 4276).

3. Line 121: The EDX profile was not provided.

Response: Thanks for the careful reading. The EDX data are added in Figure S2 (Supporting Information) in revision and related sentences of revised manuscript are highlighted (lines 14-15, page 5).

Figure S2. The EDX profile of (a) fresh $\text{Mo}_3\text{S}_4@\text{NaZSM-5}$ and (b) spent sample obtained from the selected yellow square in Figure 2 (a-d).

4. Line 144: This can not confirm the external dispersion of MoS_2 on NaZSM-5 zeolite in $\text{MoS}_x/\text{NaZSM-5}$.

Response: Thanks for the comment. The HRTEM characterization of $\text{MoS}_x/\text{NaZSM-5}$ has been added as Figure S3 in revision (Supporting Information). It clearly shows that the layered MoS_2 in size of ~ 10 nm, much larger than ZSM-5 channels, are supported on the external surface of NaZSM-5 zeolite in $\text{MoS}_x/\text{NaZSM-5}$. Revisions have been made accordingly and highlighted (lines 16-20, page 5 and lines 4-6, page 6).

Figure S3. The HRTEM image of MoS_x/NaZSM-5 catalyst.

5. Line 145: How is the size of [Mo₃S₄]ⁿ⁺ clusters in Mo₃S₄@NaZSM-5 calculated?

Response: By the theoretically optimized configuration, the size [Mo₃S₄]ⁿ⁺ is ~0.4 nm.

6. Figure 2k-2l: the shift of binding energy in XPS spectra is generally related to the electron transfer from one element to another. Why do S and Mo both move to lower binding energy after reaction?

Response: It is a good question. During reaction, some fragments of hydrocarbons can be detected by in-situ IR measurements, which may cause the catalyst in more reduced state. Thus, both the binding energies of molybdenum and Sulphur move to lower values. In addition, by the Figure 2j, the coordination of oxygen from zeolitic framework seems decreased after reaction, maybe caused by the tiny movement of the Mo₃S₄ clusters in channel of ZSM-5, also results in the binding energy of Mo and S moving to lower positions. (lines 15-17 and 20-23, page 6).

7. Figure 2k-2l: XPS is a surface analysis method. If the MoS_x cluster was embedded into the pores of zeolite, it should be very hard to detect the S and Mo elements. Why is the intensity of S and Mo peaks in Mo₃S₄@NaZSM-5 so strong?

Response: Thanks for the thoughtful question. Indeed, the signals are very weak and we spent many instrument times to collect the X-ray photo electrons from the samples to modify the ratio of signal to noise. The PHI5000 Versa Probe XPS system we used also in a good status. Even though, since Mo₃S₄ clusters are embedded in the pores of ZSM-5 zeolite in Mo₃S₄@NaZSM-5, the intensity

of Mo and S peaks in $\text{Mo}_3\text{S}_4@\text{NaZSM-5}$ is significantly lower than those in $\text{MoS}_x/\text{NaZSM-5}$ (Figure 2k-2l). As to the slightly higher signal strength of Mo and S in $\text{Mo}_3\text{S}_4@\text{NaZSM-5}$ than that of $\text{MoS}_2+\text{NaZSM-5}$ is due to that MoS_2 in $\text{MoS}_2+\text{NaZSM-5}$ is too bulky and its dispersion is too low.

8. Line 176: XPS spectrum of MoO_3 should be added.

Response: Yes, XPS spectrum of MoO_3 are added in Figure 2k. Thanks.

Figure 2k. Mo 3d XPS spectrum of MoO_3 , commercial $\text{MoS}_2+\text{NaZSM-5}$, $\text{MoS}_x/\text{NaZSM-5}$, $\text{Mo}_3\text{S}_4@\text{NaZSM-5}$, and Spent $\text{Mo}_3\text{S}_4@\text{NaZSM-5}$.

9. Table S1: How is the S/Mo ratio measured? The ESCAR characterization method is not explained in the section of Characterization techniques.

Response: Thanks for the question and reminding. The acronym “ESCA” means “Electron Spectroscopy for Chemical Analysis” and was wrong written as “ESCAR” in the original manuscript. Thanks very much for reminding. The explanation is added to the Table S1 in revision (Supporting Information). And the characterization method is added in Characterization techniques in revision and related sentences of revised manuscript are highlighted (lines 15-18, page 17). Thanks again for your very careful reading.

10. The calculation formula for CH_3OH selectivity was unreasonable.

Response: Thanks for the careful question. The description of original manuscript is not so strict.

It has been revised and highlighted in revised manuscript. In current work, only CH₄, CO and CH₃OH are detected products. The sum of their selectivity is 100%. Revision has been made accordingly. (lines 37-39, page 16 and line 1, page17)

11. Table 3: there seems to be something wrong with the data of product selectivity. The O element during reaction is imbalance.

Response: Thanks a lot for the careful reading. Indeed, the data of selectivity listed in 1,3 rows of table 3 are wrong. During revision, we have retested the catalytic data and have found the cause of the mistake. The Table 3 is corrected in the revised manuscript. Thanks again. (page 14)

12. Line 321: why did the hydrocarbons distribution over Mo₃S₄@HZSM-5 deviate far from the ASF model?

Response: Thanks for the thoughtful question. The hydrocarbons distribution of ASF model by CO hydrogenation is obtained through the F-T synthesis. In general, F-T synthesis firstly involves the dissociation of CO and then generates surface CH_x (x=1-3). The continuous methylene insertion causes chain growth of surface species C_nH_m. C_nH_m species are hydrogenated or dehydrogenated to paraffin or olefin products. The random desorption of the surface species leads to product distribution follows ASF rule. In current investigation with Mo₃S₄@HZSM-5 catalyst, the primary product should be methanol which was transformed into light olefins at proton sites of HZSM-5 zeolite and then most olefins are further hydrogenated to alkanes over Mo₃S₄ clusters. CO activation and C-C bonding take place at different active sites, which should be viewed as a cascade route of reaction, and then the products deviate from ASF distribution.

The authors especially thank the reviewer #2 for his/her careful reading and profound scientific questions, which are very helpful to modify the scientific quality of the manuscript.

Reviewer #3 (Remarks to the Author): The manuscript “Mo₃S₄@ZSM-5 for Highly Efficient Hydrogenation of CO_x to Methanol and Hydrocarbons under Mild Reaction Conditions”, submitted by Liu et al, is dedicated to the important research on the direct conversion of CO₂ and CO into valuable chemicals and contains quite a number of interesting results. Through appropriate modification of the Mo₃S₄@ZSM-5 catalyst, both CO₂ and CO can be converted into methanol and low hydrocarbons in high selectivity. Various important characterization means are explained and

presented in good manners. This reviewer recommends the acceptance of the work for publication by Nature Communications after minor revisions, considering the following points.

1. Keywords. Mo₃S₄ cluster should be written as [Mo₃S₄]ⁿ⁺ cluster.

Response: Thanks for the careful reading. Revisions have been made accordingly and marked with yellow shading. (line 3, page 2)

2. Why the ZSM-5 zeolite was used in this work and have the authors explored the influence of Si/Al on the catalyst? Whether other types of zeolites are feasible?

Response: Thanks for the thoughtful questions. We have been impressed for a long time by unique catalytic properties of ZSM-5 zeolite with its intracrystalline tough clusters of metal compounds, such as MoC_x@ZSM-5 (*J. Phys. Chem. B* 2001, 105, 506), WC_x@ZSM-5 (*J. Phys. Chem. B* 2001, 105, 3928), MoN_x@ZSM-5 (*ChemCatChem* 2010, 2, 167) and Pt_x@ZSM-5 (*ACS Catal.* 2015, 5, 6893). The strong regulatory effect of ZSM-5 on the confined clusters and the synergistic interaction between them on catalytic reaction leads to their catalytic property unpredictable and extremely interesting. We have explored the influence of Si/Al ratio on the synthesis of Mo₃S₄@HZSM-5 catalyst and found that HZSM-5 zeolite with Si/Al ratio of 15-20 was the most suitable for the synthesis of Mo₃S₄@HZSM-5 catalyst. SSZ-39, SAPO-34 and Beta zeolites have been also investigated to regulate molybdenum sulfide clusters and they basically show remarkable differences for the reaction. We will successively report these results. Thanks again.

3. Catalyst preparation. Can NaZSM-5 zeolite be directly used to synthesize Mo₃S₄@NaZSM-5 by solid exchange method?

Response: That is a good question. By experimental, NaZSM-5 zeolite cannot be directly used to synthesize Mo₃S₄@NaZSM-5 by solid exchange method because there is no exchange site with MoO₃ in NaZSM-5 zeolite. It seems charged NaMoO₄ clusters cannot be formed in the ZSM-5 zeolite channel.

4. Figure S4. The cross-sectional diameter of 2,3-dimethylnitrobenzene is slightly larger than the channel diameters of ZSM-5, but the Mo₃S₄@HZSM-5 catalyst still shows certain catalytic activity during the hydrogenation of 2,3-dimethylnitrobenzene. Is this caused by part of MoS_x not entering the micropores of ZSM-5 zeolite?

Response: Thanks for the question. Yes, the cross-sectional diameter of 2,3-dimethylnitrobenzene is indeed slightly larger than the channel diameters of ZSM-5 zeolite, which makes it difficult to enter the micropores of ZSM-5 zeolite. The small activity for hydrogenation of 2,3-dimethylnitrobenzene over $\text{Mo}_3\text{S}_4@\text{HZSM-5}$ catalyst would be caused by small part of MoS_x at the external surface of ZSM-5 zeolite or by some MoS_x clusters at cages of ZSM-5 near the surface.

5. Why do $\text{Mo}_3\text{S}_4@\text{NaZSM-5}$ and $\text{Mo}_3\text{S}_4@\text{HZSM-5}$ show excellent performance in CO_2 and CO hydrogenation reactions, respectively? How alkali metals affect the electronic structure of molybdenum species? What is the nature of the active site?

Response: Thanks for the thoughtful questions. As we reported previously, the MoN_x clusters in $\text{MoN}_x@\text{HZSM-5}$ (*ChemCatChem* 2010, 2, 167) and the ruthenium clusters in NaX zeolite (*Catal. Sci. & Tech.* 2018, 8, 6384) are affected significantly by the intracrystal dielectric field of the zeolite, electronic structure, such as Fermi level of the clusters, and then catalytic property. In addition, zeolite structures are also very influential. SSZ-39, SAPO-34 and Beta zeolites have been also investigated to regulate molybdenum sulfide clusters and they basically show remarkable differences for the reaction. Such catalysts can show many unexpected catalytic properties.

6. As far as I know, the ZSM-5 is often used to synthesize C_{5+} hydrocarbons in syngas conversion. However, the selectivity of $\text{C}_2\text{-C}_3$ hydrocarbons is more than 98% in organics on $\text{Mo}_3\text{S}_4@\text{HZSM-5}$. Why?

Response: Thanks for the careful reading. Firstly, the reaction temperature is relatively low and the size of $[\text{Mo}_3\text{S}_4]^{n+}$ cluster is too small to offer more sites nearby for reaction of methylene insertion, which causes less hydrocarbons in long chains. Secondly, the possible cracking reaction of products at the proton site of the zeolite also inhibit the generation of C_{5+} hydrocarbons.

REVIEWERS' COMMENTS

Reviewer #1 (Remarks to the Author):

The authors have revised the manuscript according to the reviewers' concerns. I consider this manuscript publishable in Nature Communications.

Reviewer #2 (Remarks to the Author):

The questions have been well addressed in the revised paper. This paper can be published.

Reviewer #3 (Remarks to the Author):

The authors have addressed all my issues. It can be accepted after the revision.